# Ag_2_O and NiO Decorated CuFe_2_O_4_ with Enhanced Photocatalytic Performance to Improve the Degradation Efficiency of Methylene Blue

**DOI:** 10.3390/ma13214760

**Published:** 2020-10-25

**Authors:** Lu Liu, Nan Hu, Yonglei An, Xingyuan Du, Xiao Zhang, Yan Li, Yan Zeng, Zheng Cui

**Affiliations:** 1School of Energy and Power Engineering, Changchun Institute of Technology, Changchun 130012, China; zhangx049@nenu.edu.cn (X.Z.); moon471285743@126.com (Y.L.); zengyan@ccit.edu.cn (Y.Z.); 2Jilin Province S&T Innovation Center for Physical Simulation and Security of Water Resources and Electric Power Engineering, Changchun Institute of Technology, Changchun 130012, China; 3Key Laboratory of Groundwater Resources and Environment, Jilin University, Ministry of Education, Changchun 130021, China; anyonglei85@jlu.edu.cn (Y.A.); duxy19@mails.jlu.edu.cn (X.D.); 4Jilin Provincial Key Laboratory of Water Resources and Environment, Jilin University, Changchun 130021, China; 5State Key Laboratory of Superhard Materials, College of Physics, Jilin University, Changchun 130012, China; cuizheng18@mails.jlu.edu.cn

**Keywords:** CuFe_2_O_4_, Ag_2_O, NiO, photocatalysis, methylene blue

## Abstract

Dye wastewater is a serious threat to human health and life. It is an important task for researchers to treat it efficiently. Among many treatment methods, the photo-Fenton method can rapidly degrade organic pollutants. In this study, a ternary photocatalyst, Ag_2_O-NiO/CuFe_2_O_4_, was prepared and applied for a photo-Fenton reaction to degrade methylene blue (MB). MB had the best degradation effect when 10 mg of the catalyst were used in an 80 mL reaction system for measurement. The degradation rate of MB was up to 96.67% in 60 min with a high degradation rate constant $k=5.67{\times 10}^{-2}\min^{-1}$. The total organic carbon (TOC) degradation rate was 78.64% with a TOC degradation rate constant of $k=2.57{\times 10}^{-2}\min^{-1}$. Therefore, this study fully proves that Ag_2_O-NiO/CuFe_2_O_4_ can catalyze the photo-Fenton reaction and effectively degrade MB.

## 1. Introduction

With the rapid development of global industry, the problem of water pollution has become more and more severe. Organic dyes based on heterocyclic aromatic/aromatic-azo compounds are highly toxic, carcinogenic, and difficult to degrade [1,2]. Methylene blue (MB) mainly comes from the printing and dyeing industry [3]. MB can prevent sunlight from entering water and therefore affects the growth of aquatic organisms [4]. Furthermore, MB can cause eye burns and may even lead to permanent damage. If MB is taken into the body, it will cause a burning sensation and vomiting. Therefore, the treatment of dye wastewater has become an urgent problem. Unfortunately, traditional wastewater treatment processes (microbial degradation) cannot efficiently decompose MB. 

In recent years, researchers have proposed many new technologies, such as adsorption, condensation, photocatalysis, electrocatalysis, and the photo-Fenton process [5,6,7,8]. Solar energy is an inexhaustible clean energy source that can be used to degrade MB and other dyes. Due to the economics of solar energy, photocatalysis is considered one of the most promising methods. Many semiconductor materials can decompose toxic organic compounds into non-toxic inorganic compounds under light irradiation. Various semiconductors have been developed as photocatalysts, including CuFe_2_O_4_, TiO_2_, CdS, Bi_2_O_3_, Ag_2_O, NiO, and *g*-C_3_N_4_ and so on [9,10,11,12]. Copper ions and iron ions can participate in the photo-Fenton reaction [13]. Therefore, CuFe_2_O_4_ can also trigger the photo-Fenton reaction.

Among many photocatalysts, CuFe_2_O_4_ has attracted wide attention because of its suitable energy band structure, high chemical stability, thermal stability and low toxicity [14]. In addition, CuFe_2_O_4_ has magnetic properties that makes it easy to recycle [15,16]. It will not cause secondary pollution and can be widely used in heterogeneous catalysis. CuFe_2_O_4_ has a relatively narrow band gap [17] that can absorb visible light. In order to increase its absorption rate of visible light, it is often prepared into a micro-nano structure [18,19]. The surface of the structure produces defect energy levels and surface energy levels during the preparation process, thereby providing a recombination center for photo-generated electron-holes [20,21]. Then, the lifetime of electrons and holes is reduced, i.e., the photocatalytic efficiency is reduced. Therefore, a single transition metal oxide used for the photocatalytic degradation of MB cannot meet the needs of practical applications. At present, researchers have made many efforts to improve the photocatalytic properties of materials, such as noble metal loading, metal doping, and coupling with other semiconductors (MoS_2_, Co_3_O_4_, NiO, WO_3_, etc.) [9,22,23,24]. The latest literature has reported that mixed transition metal oxides (CuO/Fe_3_O_4_, CuO/MnFe_2_O_4_, *γ*-Fe_2_O_3_/Mn_3_O_4_, and CoO-CuO) not only have excellent catalytic performance but also maintain excellent stability [24,25,26]. Therefore, the combination of CuFe_2_O_4_ and other semiconductors would increase the photo-Fenton reaction rate.

Ag_2_O and NiO are often used in photocatalysis research. Though it is a narrow-band gap semiconductor, Ag_2_O is rarely used as the main catalyst, but it is used as a co-catalyst due to the instability of its photocatalytic reaction and its high carrier recombination rate [23,27]. NiO is a wide band gap semiconductor with important electronic, chemical, and electrical properties [28]. NiO can only absorb ultraviolet light, so its utilization rate of sunlight is low. It is the best choice to combine NiO with other semiconductor materials to improve its photocatalytic activity. 

Based on above introduction, the hydrothermal method and calcination were used to prepare an Ag_2_O-NiO/CuFe_2_O_4_ ternary catalyst, which showed an excellent photocatalytic performance under sunlight. Within 60 min of photocatalytic degradation in photo-Fenton system, the removal rate of MB was 96.67%. The degradation mechanism of MB in this study was analyzed according to its characterization and catalytic performance.

## 2. Material and Methods

### 2.1. Chemicals

Copper sulfate pentahydrate (CuSO_4_·5H_2_O, analytical reagent (AR)), ferric chloride hexahydrate (FeCl_3_·6H_2_O, AR), nickel acetate tetrahydrate (C_4_H_6_NiO_4_·4H_2_O, AR), silver sulfate (Ag_2_SO_4_, AR), sodium hydroxide (NaOH, AR), methylene blue (MB, AR), 30.0% hydrogen peroxide (H_2_O_2_, AR), absolute ethanol (EtOH, C_2_H_5_OH, AR), and *t*-butanol (TBA, (CH_3_)_3_COH, AR) were purchased from Sinopharm Chemical Reagent (Shanghai, China) and were used as received without further purification.

### 2.2. Synthesis and Characterization of Ag_2_O-CuFe_2_O_4_/NiO

In 200 mL deionized water, 3.210 g CuSO_4_·5H_2_O and 2.705 g FeCl_3_·6H_2_O were fully dissolved under magnetic stirring; this is marked as solution A. Additionally, a 100 mL NaOH solution was prepared with a final pH of 9; this is marked as solution B. We dropwise added solution A to solution B under magnetic stirring, and we kept the pH of the mixture at 9 from beginning to end by adding 1.0 M NaOH, thus generating a black suspension. Then, the suspension was evaporated and concentrated to 60 mL at 60 °C under magnetic stirring. The 60 mL concentrated suspension was poured into a stainless steel reactor and placed in a 100 °C oven for 10 h. Subsequently, the concentrated suspension was centrifuged and dried to obtain a black powder, which is denoted as powder C. To remove undesired soluble ions such as Na^+^, SO_4_^2−^ and Cl^−^, powder C was washed with deionized water and then centrifuged three times; then, precipitate powder C was dried in a 60 °C oven overnight. Then, 0.117 g of Ag_2_SO_4_ and 0.031 g of C_4_H_6_O_4_Ni·4H_2_O were dissolved in 20 mL of deionized water, and all of black powder C was poured in; this is denoted as mixture M. Mixture M was tempestuously stirred at room temperature for 2 h to make powder C fully and uniformly contact Ag^+^ and Ni^2+^. Then, mixture M was dried at 60 °C under magnetic stirring until the deionized water completely evaporated. We then put the residual solid mixture M into a muffle furnace for high-temperature calcination in an air atmosphere. The detailed process of sample preparation is shown in Figure 1. Appendix A shows the MB degradation with the samples calcined at different temperatures. It can be found from Appendix A that the sample calcined at 650 °C had the best degradation effect of MB, so the calcination temperature of all samples used in this study was 650 °C and the sample was coded as Ag_2_O-NiO/CuFe_2_O_4_. The molar ratio among Ag_2_O, NiO, and CuFe_2_O_4_ was calculated as 3:1:40 in Ag_2_O-NiO/CuFe_2_O_4_. For control experiments, a sample without the addition of Ag_2_SO_4_ and C_4_H_6_O_4_Ni·4H_2_O was also prepared by directly calcining powder C; this is coded as CuFe_2_O_4_.

The phase composition and crystal structure of Ag_2_O-NiO/CuFe_2_O_4_ were analyzed by transmission electron microscopy (TEM) with a JEM2100 (JEOL, Tokyo, Japan) and X-ray diffraction (XRD) with an XRD-6000 (Shimadzu, Kyoto, Japan) with the radiation (λ = 1.5406 Å) in the range of 2θ from 10° to 70° and a 1°/min scanning rate.

### 2.3. Photodegradation Experiments

The photocatalytic performance of Ag_2_O-NiO/CuFe_2_O_4_ was evaluated with MB. A 500 W Xe lamp (GXZ500, Shanghai Jiguang special lighting electric appliance factory, Shanghai, China) was used as the simulated sunlight. The distance from the lamp to the MB suspension liquid level was 15 cm. As is typical, 10.0 mg of Ag_2_O-NiO/CuFe_2_O_4_ and 100 μL of H_2_O_2_ were simultaneously added into an aqueous MB solution (80 mL, 10 mg/L), and then the MB photocatalytic degradation experiment was carried out under the irradiation of the Xe lamp. At a certain time interval, a 3 mL suspension was withdrawn from the reaction and subjected to centrifugation. In addition, the same reaction mixture was magnetically stirred in the dark to compare the results irradiated by the Xe lamp. The MB concentration was determined by a UV-VIS spectrophotometer (UV-1240 Shimadzu, Kyoto, Japan) at the wavelength of 664 nm. The total organic carbon (TOC, mg/L) of the MB solution was determined by a TOC analyzer (SSM-5000A Shimadzu, Kyoto, Japan) during the photocatalytic degradation process. To reveal the photocatalytic mechanism, typical radical scavengers such as EtOH and TBA were used in reactions to verify the active species. We used the same method to analyze the photo-Fenton performance of CuFe_2_O_4_, NiO/CuFe_2_O_4_, andAg_2_O/CuFe_2_O_4_ as the controls. The effects of NiO/CuFe_2_O_4_ and Ag_2_O/CuFe_2_O_4_ on MB degradation are shown in Appendix A.

The degradation efficiency of MB can be calculated by Equation (1) [29]:Degradation (%) = (1 − C/C_0_) × 100%(1)
where C and C_0_ represent the concentration of MB at time *t* and the pristine concentration of MB, respectively.

In this study, Equation (2) [30] was used to analyze the pseudo-first-order dynamics.
−ln(*C/C_0_*) = *kt*(2)
where *k* is the apparent reaction rate constant (min^−1^).

## 3. Results and Discussion

### 3.1. Characterization of the Synthesized Ag_2_O-NiO/CuFe_2_O_4_

The phase composition and crystal structure of the sample studied by XRD are shown in Figure 2. The XRD data proved that the material contained three components: Ag_2_O, NiO, and CuFe_2_O_4_. The 18.55°, 30.18°, 35.65°, 44.70°, and 65.01° diffraction peaks appeared in the CuFe_2_O_4_ and Ag_2_O-NiO/CuFe_2_O_4_ samples. They corresponded to the (101), (112), (211), (202), (220), and (400) crystal planes of CuFe_2_O_4_ (PDF card No.72-1174), respectively. The 32.74° and 53.75° diffraction peaks in Ag_2_O-NiO/CuFe_2_O_4_ corresponded to the (111) and (220) crystal planes of Ag_2_O (PDF card No.12-793), respectively. The 37.29° and 62.67° diffraction peaks in Ag_2_O-NiO/CuFe_2_O_4_ corresponded to the (101) and (110) crystal planes of the NiO phase (PDF card No.44-1159), respectively. These results could convincingly verify the existence of Ag_2_O and NiO in Ag_2_O-NiO/CuFe_2_O_4_. Therefore, it was proven that Ag_2_O, NiO, and CuFe_2_O_4_ were the main components in the sample and the as-prepared Ag_2_O-NiO/CuFe_2_O_4_ was synthesized successfully.

The morphology of Ag_2_O-NiO/CuFe_2_O_4_ was characterized and analyzed by TEM, as shown in Figure 3. In Figure 3b, the crystal lattice fringe of the sample can be clearly seen, which fully proves that the sample had an excellent crystallinity within a certain size range that was conducive to carrier transmission. Figure 3c–e shows the enlargements of the corresponding square areas in Figure 3b. The measured lattice plane spacings were 0.48, 0.249, and 0.205 nm, corresponding to the (101), (211), and (220) crystal planes of CuFe_2_O_4_ (PDF card No.72-1174), respectively, which was consistent with the results of XRD. Due to the small proportion of Ag^+^ and Ni^2+^ doping, Ag_2_O and NiO nanoparticles were not found in the field of TEM.

### 3.2. MB Degradation

The catalytic activity of synthetic nano-material Ag_2_O-NiO/CuFe_2_O_4_ was analyzed by observing the degradation of MB in the water phase under simulated sunlight irradiation. The degradation efficiency of MB with different catalysts is shown in Figure 4. In a typical experiment, the solid catalyst dosage was 10 mg and the H_2_O_2_ dosage was 100 µL, both of which were simultaneously added to an 80 mL MB aqueous solution. Figure 4a shows the degradation of MB by Ag_2_O-NiO/CuFe_2_O_4_/H_2_O_2_ in the dark. Meanwhile, the MB degradation by CuFe_2_O_4_, CuFe_2_O_4_/H_2_O_2_, H_2_O_2_, and Ag_2_O-NiO/CuFe_2_O_4_ were also compared. The MB removal in each curve decreased within 60 min, and the degradation efficiency of Ag_2_O-NiO/CuFe_2_O_4_/H_2_O_2_ was the highest. However, the removal rate of MB was only 8.60%, and the degradation rate constant was $k=1.50{\times 10}^{-3}\min^{-1}$. The removal rate of CuFe_2_O_4_/H_2_O_2_ on MB was 7.30%, and the degradation rate constant was $k=3.34{\times 10}^{-4}\min^{-1}$. The degradation efficiency of pure H_2_O_2_ was slightly lower than that of Ag_2_O-NiO/CuFe_2_O_4_/H_2_O_2_ and CuFe_2_O_4_/H_2_O_2_. However, in the comparative experiment without H_2_O_2_, the MB removal rates of Ag_2_O-NiO/CuFe_2_O_4_ and CuFe_2_O_4_ were only 1.98% and 1.04%, respectively. Therefore, it can be seen from these studies that Ag_2_O-NiO/CuFe_2_O_4_ and CuFe_2_O_4_ decreased the concentration of MB in the dark due to the small amount of adsorption on the surface of nanomaterials. H_2_O_2_ could degrade dyes, but the degradation efficiency was low. Figure 4c depicts the MB degradation under simulated sunlight irradiation. Figure 4d shows that Ag_2_O-NiO/CuFe_2_O_4_/H_2_O_2_ had the highest pseudo-first-rate kinetic constant. Ag_2_O-NiO/CuFe_2_O_4_ and CuFe_2_O_4_ could catalyze the degradation of MB with H_2_O_2_, the MB removal rates were as high as 96.67% and 57.88%, respectively, and the corresponding degradation rate constants were $k=5.67{\times 10}^{-2}$ and $k=1.44{\times 10}^{-2}\min^{-1}$, respectively. With the assistance of Ag_2_O-NiO/CuFe_2_O_4_, the MB degradation rate constant with H_2_O_2_ under simulated sunlight irradiation was two orders of magnitude higher than that in the dark. Therefore, it was proven that we had successfully synthesized a photocatalytic material that efficiently degraded MB. In CuFe_2_O_4_, the Cu valence is +2 and the Fe valence is +3. In this study, Fe^3+^ was reduced to Fe^2+^ by photoelectrons. Cu^2+^ and Fe^2+^ can react with H_2_O_2_ to produce OH, which can efficiently degrade MB. This process is called the Fenton reaction. Under solar irradiation, the catalytic effect of the catalyst is improved, and this process is called the photo-Fenton reaction. The degradation of MB by Ag_2_O-NiO/CuFe_2_O_4_ and H_2_O_2_ under simulated sunlight irradiation was a photo-Fenton reaction.

In order to investigate the reuse potential of our photo-Fenton catalyst, Ag_2_O-NiO/CuFe_2_O_4_ was collected after each degradation experiment via centrifugation. After deionized water washing and drying at 60 °C, Ag_2_O-NiO/CuFe_2_O_4_ was reused as a reactive catalyst for MB degradation under the same experimental conditions, and the results are shown in Figure 4e. The photocatalyst used for the fifth time could still efficiently degrade MB in the aqueous phase, and the removal rate of MB was still as high as 94.02%.

The total organic carbon (TOC) of the MB solution was determined during the experiment. The TOC removal rate and the removal rate constant of Ag_2_O-NiO/CuFe_2_O_4_ were 78.64% and $k=2.57{\times 10}^{-2}\min^{-1}$ within 60 min, respectively, as shown in Figure 5. The TOC results suggested that Ag_2_O-NiO/CuFe_2_O_4_/H_2_O_2_ could effectively mineralize MB under solar irradiation.

By optimizing the amount of catalyst, the unnecessary use of a photocatalyst in degradation experiments could be minimized. For this purpose, the 10 mg/L MB was degraded for 60 min using 1, 5, 10, 20, and 50 mg of catalysts mixed with 100 µL of H_2_O_2_ in an 80 mL MB solution. The results are shown in Figure 6a,b. It was found that increasing the amount of catalyst from 1 to 50 mg could cause the MB removal rate to increase from 86.06% to 96.67% and then decrease to 81.15%. When the dosage of Ag_2_O-NiO/CuFe_2_O_4_ was 10 mg, the catalytic efficiency was the best. When the Ag_2_O-NiO/CuFe_2_O_4_ dosage was less than 10 mg, the catalytic efficiency showed a positive correlation with the catalyst dosage. Then, with the further increase of the catalyst dosage, the MB degradation efficiency significantly decreased. When the dosage of catalyst was small, all reaction sites could fully absorb sunlight and participate in the reaction together with H_2_O_2_ until the dosage of the catalyst reached the optimal value. When the dosage of catalyst excessively increased, the light could not penetrate into the dye solution due to excessive scattering of the material. Thus, the MB degradation in the internal solution was hindered, so the overall degradation effect was poor. When the dosage of Ag_2_O-NiO/CuFe_2_O_4_ was 10 mg, the degradation efficiency of MB was positively associated with the dosage of H_2_O_2_, as shown Appendix A. However, when the dosage of H_2_O_2_ was greater than 100 µL, the degradation efficiency changed slowly, so 100 µL was the cost-effective dosage for H_2_O_2_.

### 3.3. Enhanced Mechanism of Ag_2_O-NiO/CuFe_2_O_4_ on Photocatalytic Activity

In order to study the role of ·OH in the MB degradation process, EtOH and TBA were used as ·OH quenchers. Then, we studied their effect on MB degradation, as shown in Figure 7. Both EtOH and TBA inhibited MB degradation. When 10 mL of EtOH and 10 mL of TBA were added, the degradation rates of MB were 0.44% and 0.64%, respectively. These data fully proved that the increase of ·OH produced by the photo-Fenton reaction in this study led to an increase of the MB removal rate.

Ag_2_O-NiO/CuFe_2_O_4_ showed an excellent photocatalytic performance. Ag_2_O-NiO/CuFe_2_O_4_ was much more efficient in degrading MB than CuFe_2_O_4_. CuFe_2_O_4_ was a typical photocatalytic semiconductor material with band gap width E_g_ = 1.43 eV, a conduction band bottom position E_c_ = −1.12 V, and a valence band top position E_v_ = 0.31 V [31]. Under the sunlight irradiation, electrons in the valence band of CuFe_2_O_4_ could absorb the photons (*hυ* > 1.43 eV) and jump to the conduction band. Photogenerated electrons have reductive properties [32]. Thus, the ferric iron Fe(III) in CuFe_2_O_4_ can be reduced to Fe(II) via a ligand-to-metal charge transfer (LMCT) mechanism. As such, Fe(II) and Cu(II) all can catalytically decompose H_2_O_2_ into OH, which is the key active species to degrade MB dye [33]. Here, MB was adsorbed onto the surface of photocatalytic material due to hydrogen bonding, π–π interactions, surface complexation, electrostatic interactions, and chemisorption. Then, the Fe(III) and Cu(III) generated by H_2_O_2_ oxidation could be reduced into Fe(II) and Cu(II) by photo-induced electrons, but the CuFe_2_O_4_ would show an oxidation state because it lacked electron, thus resulting in a low photo-Fenton efficiency. The photo-Fenton reaction process sponsored by CuFe_2_O_4_ can be described by Equations (3)–(6) [13].

Ag_2_O and NiO are commonly used as photocatalysts. Ag_2_O has a narrow band gap, E_g_ = 1.51 eV, and electron-hole pairs can be generated by absorbing visible light [13]. NiO has a wide band gap width of 3.63 eV. The bottom of NiO conduction band is E_c_ = −0.345 V, and the top of the valence band is E_v_ = 3.285 V [34]. A ternary catalyst composed of Ag_2_O, NiO, and CuFe_2_O_4_ can greatly improve photocatalytic efficiency.
(3)$${\mathrm{CuFe}}_{2}O_{4}\equiv \mathrm{Fe}(\mathrm{III})\overset{hv}{\to }{(h^{+})\mathrm{CuFe}}_{2}O_{4}\equiv \mathrm{Fe}(\mathrm{II})$$
(4)$$(h^{+}{)\mathrm{CuFe}}_{2}O_{4}\equiv \mathrm{Fe}(\mathrm{II})+H_{2}O_{2}\to {(h}^{+}{)\mathrm{CuFe}}_{2}O_{4}\equiv \mathrm{Fe}(\mathrm{III})+\cdot \mathrm{OH}+{\mathrm{OH}}^{-}$$
(5)$${\mathrm{CuFe}}_{2}O_{4}\equiv \mathrm{Cu}(\mathrm{II})+H_{2}O_{2}\to {\mathrm{CuFe}}_{2}O_{4}\equiv \mathrm{Cu}(\mathrm{III})+\cdot \mathrm{OH}+{\mathrm{OH}}^{-}$$
(6)$${\mathrm{CuFe}}_{2}O_{4}\equiv \mathrm{Cu}(\mathrm{III})\overset{hv}{\to }{(h^{+})\mathrm{CuFe}}_{2}O_{4}\equiv \mathrm{Cu}(\mathrm{II})$$
(7)$${\mathrm{Ag}}_{2}O\overset{hv}{\to }{(h^{+})\mathrm{Ag}}_{2}O(e^{-})$$
(8)$${(h^{+})\mathrm{Ag}}_{2}{O(e^{-})+(h}^{+}{)\mathrm{CuFe}}_{2}O_{4}\equiv \mathrm{Fe}(\mathrm{III})\to {(h^{+})\mathrm{Ag}}_{2}{O+\mathrm{CuFe}}_{2}O_{4}\equiv \mathrm{Fe}(\mathrm{III})$$
(9)$${(h^{+})\mathrm{Ag}}_{2}{O(e^{-})+(h}^{+}{)\mathrm{CuFe}}_{2}O_{4}\equiv \mathrm{Cu}(\mathrm{II})\to {(h^{+})\mathrm{Ag}}_{2}{O+\mathrm{CuFe}}_{2}O_{4}\equiv \mathrm{Cu}(\mathrm{II})$$
(10)$${(h^{+})\mathrm{Ag}}_{2}O+\mathrm{MB}\to {\mathrm{Ag}}_{2}O+\mathrm{MB}\mathrm{oxidation}$$
(11)$$\mathrm{NiO}\overset{hv}{\to }\left(h+)\mathrm{NiO}(e^{-}\right)$$
(12)$${(h^{+})\mathrm{NiO}(e^{-})+(h}^{+}{)\mathrm{CuFe}}_{2}O_{4}\equiv \mathrm{Fe}(\mathrm{III})\to {(h^{+})\mathrm{NiO}+\mathrm{CuFe}}_{2}O_{4}\equiv \mathrm{Fe}(\mathrm{III})$$
(13)$${(h^{+})\mathrm{NiO}(e^{-})+(h}^{+}{)\mathrm{CuFe}}_{2}O_{4}\equiv \mathrm{Cu}(\mathrm{II})\to {(h^{+})\mathrm{NiO}+\mathrm{CuFe}}_{2}O_{4}\equiv \mathrm{Cu}(\mathrm{II})$$
(14)$$(h^{+})\mathrm{NiO}+{\mathrm{OH}}^{-}\to \mathrm{NiO}+\cdot \mathrm{OH}$$
(15)$$\cdot \mathrm{OH}+\mathrm{MB}\to {\mathrm{CO}}_{2}+H_{2}O+\ldots \ldots$$

Because the number of photoelectrons produced by CuFe_2_O_4_ is very limited and the Fe(III) and Cu(III) cannot be reduced in time, the photo-Fenton degradation effect of MB by the pristine CuFe_2_O_4_ reaction was very poor. Ag_2_O and NiO were able to promote the transition of photoelectrons in the ternary catalyst, as shown in Figure 8. The electrons in the valence band could jump into the conduction band under the sunlight excitation in Ag_2_O and NiO. Then, these photo-induced electrons could jump into the valence band of CuFe_2_O_4_ via a heterogeneous structure, thus promoting the effective reduction for Fe(III) and Cu(III) in CuFe_2_O_4_ and further improving the photo-Fenton efficiency. In addition, the photo-induced hole in Ag_2_O could also directly oxidatively degrade MB. Meanwhile, the photo-induced hole in NiO could oxidize OH^−^ or H_2_O to produce OH for MB degradation [35].

## 4. Conclusions

Ag_2_O-NiO/CuFe_2_O_4_ was synthesized by simple hydrothermal and calcination methods, and its morphology and structure characteristics were analyzed by TEM and XRD. The synthesized Ag_2_O-NiO/CuFe_2_O_4_ showed an excellent catalytic performance in the degradation of MB in wastewater via the photo-Fenton reaction. The mineralization rate of the 10 mg/L (80 mL) MB solution was 78.64% within 60 min with 10 mg of the Ag_2_O-NiO/CuFe_2_O_4_ photo-Fenton catalyst and 100 µL of H_2_O_2_. The novelty of this study lies in that Ag_2_O-NiO/CuFe_2_O_4_ is a novel, simple, low-cost and high-efficient photocatalyst.

## Figures and Tables

**Figure 1 materials-13-04760-f001:**
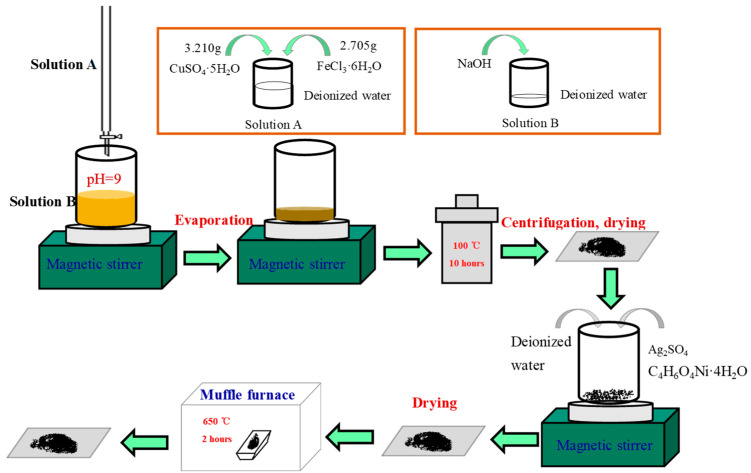
The schematic illustration of the synthetic process of Ag_2_O-NiO/CuFe_2_O_4_.

**Figure 2 materials-13-04760-f002:**
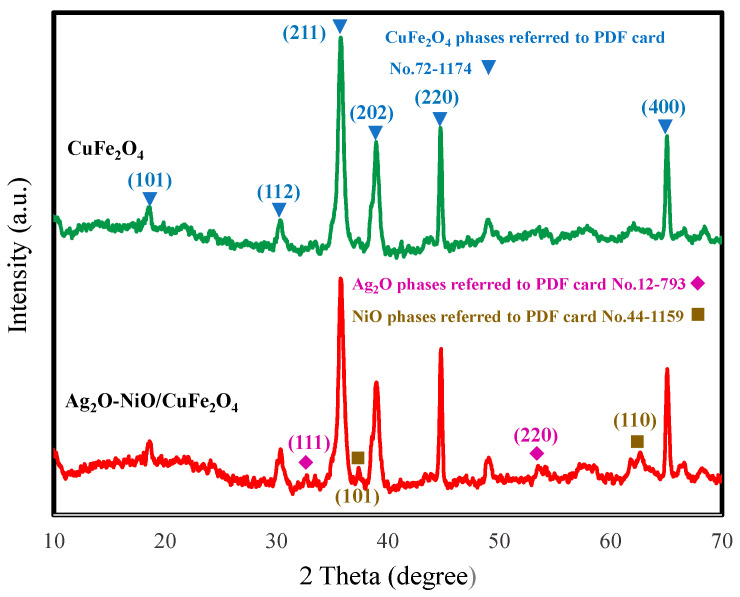
XRD patterns of CuFe_2_O_4_ and Ag_2_O-NiO/CuFe_2_O_4_.

**Figure 3 materials-13-04760-f003:**
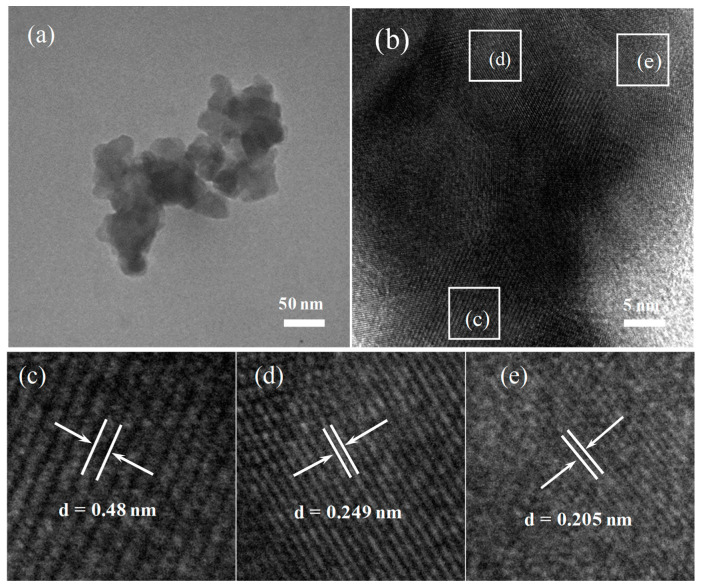
(**a**) TEM image of Ag_2_O-NiO/CuFe_2_O_4_; (**b**) High Resolution Translation Electron Microscopy (HRTEM) image of Ag_2_O-NiO/CuFe_2_O_4_; (**c**–**e**) show the enlargements of the corresponding square areas in Figure 3b.

**Figure 4 materials-13-04760-f004:**
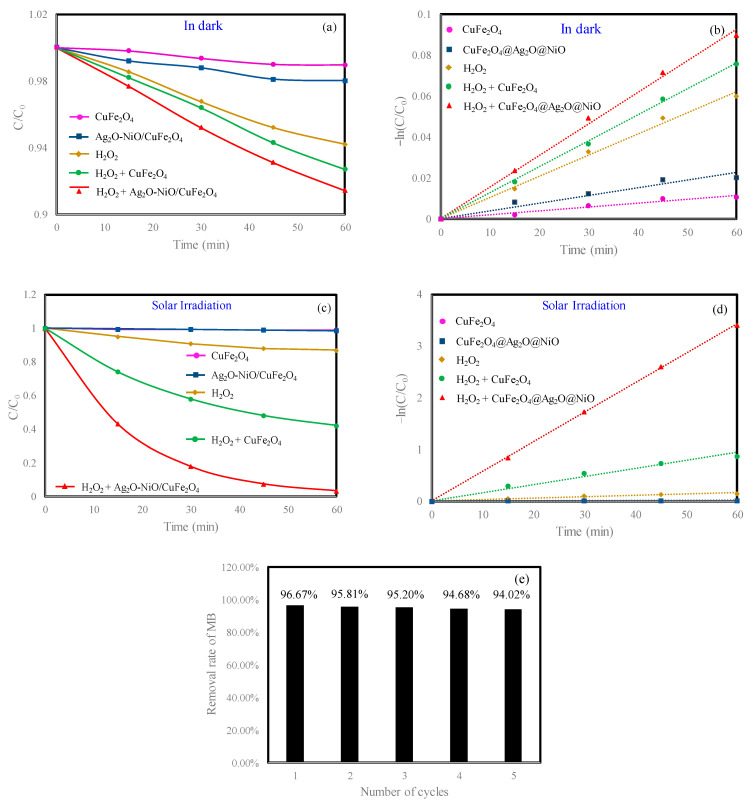
(**a**) Degradation of methylene blue (MB) with different catalysts in the dark; (**b**) the pseudo-first-order reaction kinetics for MB degradation with different catalysts in the dark; (**c**) photocatalytic degradation of MB with different composite photocatalysts under simulated solar irradiation; (**d**) the pseudo-first-order reaction kinetics for MB degradation with different composite photocatalysts under simulated solar irradiation; (**e**) percentage decrease in catalytic efficiency of Ag_2_O-NiO/CuFe_2_O_4_ for five cycles. Initial MB concentration: 10 mg/L. Initial H_2_O_2_ concentration: 416 mg/L.

**Figure 5 materials-13-04760-f005:**
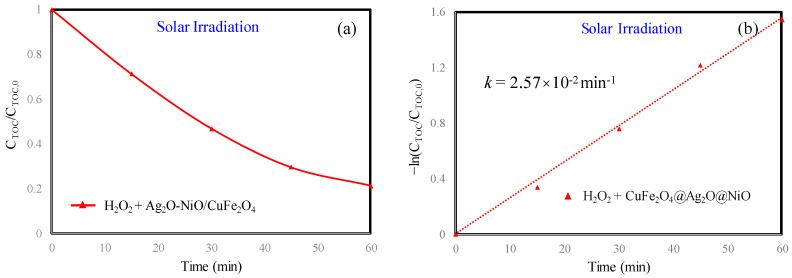
(a) Total organic carbon (TOC) variation of MB; (b) pseudo-first-order kinetic fit for TOC removal during the photo-Fenton process. Initial MB concentration: 10 mg/L. Initial H_2_O_2_ concentration: 416 mg/L.

**Figure 6 materials-13-04760-f006:**
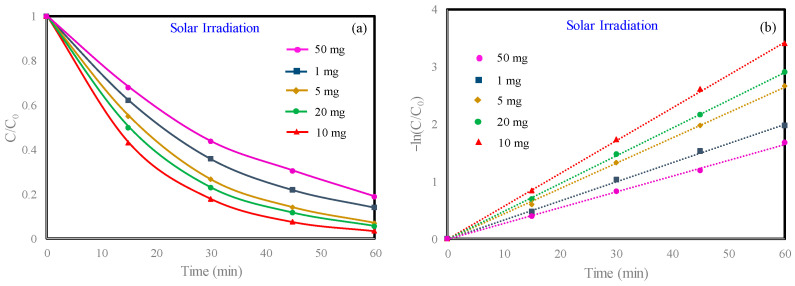
(**a**) Effect of Ag_2_O-NiO/CuFe_2_O_4_ dosage on MB dye photodegradation; (**b**) the pseudo-first-order reaction kinetics for MB dye degradation affected by different Ag_2_O-NiO/CuFe_2_O_4_ dosages. Initial MB concentration: 10 mg/L. Initial H_2_O_2_ concentration: 416 mg/L.

**Figure 7 materials-13-04760-f007:**
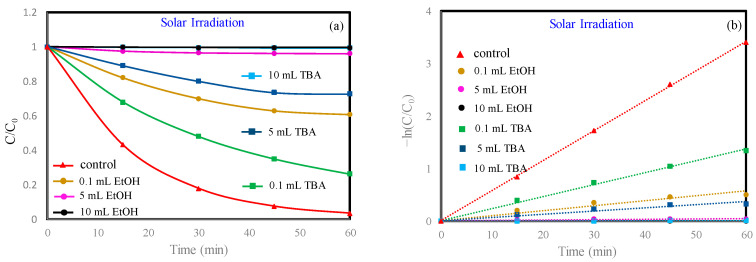
The impact of different ·OH quenchers on MB removal in a photocatalytic system. (**a**) MB degradation with different quenchers; (**b**) the pseudo-first-order reaction kinetics for MB degradation with different quenchers. Initial MB concentration: 10 mg/L. Initial H_2_O_2_ concentration: 416 mg/L.

**Figure 8 materials-13-04760-f008:**
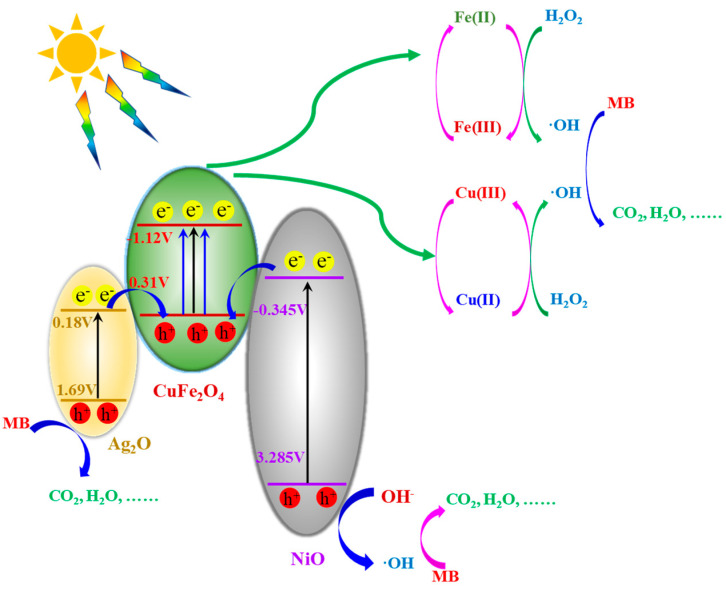
The proposed mechanism for enhanced photo-Fenton reactions with the ternary catalyst Ag_2_O-NiO/CuFe_2_O_4_.

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
