# Peer review of "Ag2O and NiO Decorated CuFe2O4 with Enhanced Photocatalytic Performance to Improve the Degradation Efficiency of Methylene Blue"

_materials, 2020, doi:10.3390/ma13214760_

Round 1

Reviewer 1 Report

The reviewer finds the manuscript very attractive where the authors present an unique, simple and low cost method with a high photocatalytic activity.

The reviewer thinks that the overall english quality of the text should be improved. For example, the reviewer has found:

Typos: 

Lines 25 and 27 "photofentun" instead of photo-Fenton.

Line 45 "participates" instead of participate.

Line 88 "power C" instead of powder C.

Repetition:

Lines 84 and 85 gradually is repeated. The reviewer suggests to rewrite it in order to avoid repetitions.

Lines 48 and 49:

When the authors wrote "CuFe2O4 has magnetism" do they mean that CuFe2O4 has magnetic properties?

On line 67 what do the authors mean with "utilization rate"? Could the authors comment on this?

Lines 77 and 78:

The authors wrote Sodium oxide and the formula provided is NaOH. Could the authors correct this?

Lines 166 and 167:

Could the authors expand on the following statement?:

"The degradation of MB by Ag2O@CuFe2O4@NiO and H2O2 is photo-Fenton reaction."

A short explanation may help the non experts to understand better the logics behind this statement.

Author Response

Dear reviewer 1,

Thank you very much for reviewing our manuscript, and giving us so good questions and advice which I think actually can improve the quality of our paper. I am also particularly grateful to you for your acknowledgement on the interesting of our paper. Now, I have revised our manuscript according to your comments, all the revisions are highlighted in red, and the detailed revision processes are described as follows:

Comment 1:

Typos: 

Lines 25 and 27 "photofentun" instead of photo-Fenton.

Line 45 "participates" instead of participate.

Line 88 "power C" instead of powder C.

Response to reviewer 1

I am sorry for my carelessness. Now, I have taken a in-depth revision of the language for our manuscript. “photofentun” has been replaced by “photo-Fenton”. “participates” has been replaced by “participate”. “power C” has been replaced by “powder C”. In addition, I have carefully checked all the English language of our manuscript.

Comment 2:

Repetition:

Lines 84 and 85 gradually is repeated. The reviewer suggests to rewrite it in order to avoid repetitions.

Response to reviewer 1

I am sorry for my carelessness. “Gradually add solution B to solution A gradually until the PH~9 under magnetic stirrer” has been modified to “Gradually add solution B to solution A until the PH~9 under magnetic stirrer” .

Comment 3:

Lines 48 and 49:

When the authors wrote "CuFe2O4 has magnetism" do they mean that CuFe2O4 has magnetic properties?

Response to reviewer 1

I am sorry for my carelessness. “CuFe2O4 has magnetism ”has been modified “CuFe2O4 has magnetic properties”.

Comment 4:

On line 67 what do the authors mean with "utilization rate"? Could the authors comment on this?

Response to reviewer 1

Think you very much for this question. The solar spectrum ranges from 380 nm to 710 nm. However, the band gap width of nickel oxide is about 3.63 eV, and it can only absorb photons greater than 3.63 eV, while those less than 3.63 eV are wasted, so the utilization rate of sunlight is low.

Comment 5:

Lines 77 and 78:

The authors wrote Sodium oxide and the formula provided is NaOH. Could the authors correct this?

Response to reviewer 1

I am sorry for my carelessness. In the experiment, sodium hydroxide was actually used. “Sodium oxide” has been modified to “sodium hydroxide”.

Comment 6:

Lines 166 and 167:

Could the authors expand on the following statement?:

"The degradation of MB by Ag2O@CuFe2O4@NiO and H2O2 is photo-Fenton reaction."

A short explanation may help the non experts to understand better the logics behind this statement.

Response to reviewer 1

Thank you very much for your advice. In this study, Fe3+ was reduced to Fe2+ by electrons. Cu2+ and Fe2+ reacted with H2O2 to produce •OH, which degraded MB. This process was called Fenton reaction. Under light irradiation, the catalytic effect of the catalyst was improved, and this process was called photo-Fenton. This explanation has been added to the corresponding position of the text.

Thank you very much again for your comments. I hope this revision is satisfied to you and our paper could be accepted soon.

Reviewer 2 Report

An interesting set of results is presented in this manuscript. Photocatalytic efficiencies are high and the materials are novel. However, several aspects require thorough revision, especially synthesis, the identity of the Ag2O and NiO components, and the mechanistic proposal, which is not totally substantiated and might involve misconceptions. Comments:

  1. Synthetic protocols are unclear.
    • Line 85: How is solvent removed at room temperature?
    • Line 90 is illegible and has to be rewritten: How are Ag and Ni components incorporated on CuFe2O4? How long is the contact in solution? At which temperature? How is the resulting solid separated? Figure 1 shows some illustration, but the information is missing in the text and the protocol cannot be reproduced.
  2. There is no evidence for photo-Fenton, and identification of Fe in different oxidation states should be attempted.
  3. Cu is not in Cu(III) state as shown in Eq. 4 and Figure 8, but as Cu(II). The derived description is thus wrong.
  4. Regarding oxidation mechanisms the authors should cite Chem. Rev. 2014, 114, 9919; Coord. Chem. Rev. 2016, 315, 1 Chem. Chem. Phys. 2014, 16, 1788.
  5. Materials with only Ag2O or NiO, that is, Ag2O/CuFe2O4 and NiO/CuFe2O4 should be prepared and tested to investigate on the different photocatalytic roles.
  6. Photocatalytic efficiency should be compared with benchmark materials such as Fe2O3, or even a non-iron one such as TiO2.
  7. The title states Ag2O and NiO are decorated onto CuFe2O4, but there is no evidence about this. Unless the different components are located, for example, by TEM-EDX elemental mapping, the morphology cannot be fully described.
  8. The material is coded as Ag2O@CuFe2O4@NiO, but “@” generally describes encapsulation of one material into another one. If Ag2O and NiO are on CuFe2O4, then another code such as Ag2O-NiO/CuFe2O4 is more suitable.
  9. Figure captions must specify all experimental conditions. For example, is MB concentration 10 mg/L in Fig. 5 and 100 mg/L in Figure 6, as mentioned in the text? Is Fig 4 data for sunlight irradiations with H2O2 or without?
  10. Lines 178-9 describes operation with H2O2, please specify.
  11. TOF must be defined and calculated.
  12. The text contains erroneous terms and typos and English is poor; major editing is needed. Examples:
    • Abstract: the terms photofentun and ternyl do not exist.
    • Experimental: what is AR? Silver nitrate or silver sulfate? Sodium hydroxide, NOT sodium oxide. Methylene blue, NOT methyl blue.
    • Line 89: powder, NOT power.
    • Line 148, 100 µL, NOT 100 mL
    • Lines 186, 198, 199, µL, NOT uL.

Author Response

Dear reviewer 2,

Thank you very much for reviewing our manuscript, and finding out so detailed  mistakes which I think actually can improve the quality of our paper. I am also particularly grateful to you for your acknowledgement on the novelty of our paper. Now, I have revised our manuscript according to your comments, all the revisions are highlighted in red, and the detailed revision processes are described as follows:

Comment 1:

Synthetic protocols are unclear.

Line 85: How is solvent removed at room temperature?

Line 90 is illegible and has to be rewritten: How are Ag and Ni components incorporated on CuFe2O4? How long is the contact in solution? At which temperature? How is the resulting solid separated? Figure 1 shows some illustration, but the information is missing in the text and the protocol cannot be reproduced.

Response to reviewer 2:

Thank you very much for this comment. To remove the unreacted solvent, the powder was repeatedly washed with deionized water and then centrifuged, at room temperature. This content has been added to section 2.2.

The preparation process of Ag2O-NiO/CuFe2O4 has been rewritten. As follow: 0.117 g Ag2SO4 and 0.031 g C4H6O4Ni・4H2O were dissolved in 20 mL deionized water, and then black powder C was poured. The mixture was thoroughly stirred at room temperature for 2 h to make the powder fully in contact with Ag1+ and Ni2+. The mixture is repeatedly washed by centrifugation and then dried.

    Comment 2:

There is no evidence for photo-Fenton, and identification of Fe in different oxidation states should be attempted.

Response to reviewer 2:

Thank you very much for your comment. XRD test results show that Ag2O-NiO/CuFe2O4 was successfully prepared. In Ag2O-NiO/CuFe2O4, Cu has +2 valence and Fe has +3 valence. The electron can reduce Fe3+ to Fe2+. Cu2+ and Fe2+ participate in the photo-Fenton reaction together.

Since the trivalent Fe was used in the experiment, the resulting material Fe was still trivalent. When electrons participated in the reduction reaction, the valence of Fe decreased.

Comment 3:

Cu is not in Cu(III) state as shown in Eq. 4 and Figure 8, but as Cu(II). The derived description is thus wrong.

Response to reviewer 2:

Thank you very much for this comment. In CuFe2O4, Cu valence is +2 and Fe valence is +3. In this study, Fe3+ was reduced to Fe2+ by electrons. Cu2+ and Fe2+ reacted with H2O2 to produce ·OH, which degraded MB. This process was called Fenton reaction. Under light irradiation, the catalytic effect of the catalyst was improved, and this process was called photo-Fenton. The Fenton reaction process involving CuFe2O4 can be analyzed by formula(3)~(7). The reaction process has been reanalysed in the text. The order of formulas has been changed.

Comment 4:

Regarding oxidation mechanisms the authors should cite Chem. Rev. 2014, 114, 9919; Coord. Chem. Rev. 2016, 315, 1 Chem. Chem. Phys. 2014, 16, 1788.

Response to reviewer 2:

Thank you very much for recommending literatures. I have carefully read these three papers and studied more photocatalysis knowledge. And above three literatures are all cited in the revised manuscript.

Comment 5:

Materials with only Ag2O or NiO, that is, Ag2O/CuFe2O4 and NiO/CuFe2O4 should be prepared and tested to investigate on the different photocatalytic roles.

Response to reviewer 2:

Thank you very much for your advice. Ag2O/CuFe2O4 and NiO/CuFe2O4 were successfully prepared and used in the experiment of photo-Fenton degradation of MB. The catalytic effect of these two materials was lower than that of Ag2O-NiO/CuFe2O4. The picture is the effect picture of Ag2O/CuFe2O4 and NiO/CuFe2O4 degrading MB, which has been added to the supporting file. Sections 2.2 and section 2.3 describe the knowledge of Ag2O/CuFe2O4 and NiO/CuFe2O4.

Figure S2. Degradation of MB with Ni/CuFe2O4 and Ag2O/CuFe2O4 under simulated solar irradiation. Experimental conditions: H2O2 100 uL, Catalyst 10 mg. The degradation rates of MB by Ni/CuFe2O4 and Ag2O/CuFe2O4 were 58.92% and 68.91% respectively, which are both higher than CuFe2O4(57.88%).

Comment 6:

Photocatalytic efficiency should be compared with benchmark materials such as Fe2O3, or even a non-iron one such as TiO2.

Response to reviewer 2:

Thank you very much for your advice. TiO2 and Fe2O3 are excellent photocatalysts. This study focuses on the mechanism of Ag2O-NiO/CuFe2O4 photo-Fenton degrade MB, so it is not compared with other catalysts. In future studies, the performance of several photocatalysts will be compared.

Comment 7:

The title states Ag2O and NiO are decorated onto CuFe2O4, but there is no evidence about this. Unless the different components are located, for example, by TEM-EDX elemental mapping, the morphology cannot be fully described.

Response to reviewer 2:

Thank you very much for your advice. I also think that TEM-EDX elemental mapping and the morphology can fully analyze samples, but during the COVID-19 epidemic, the laboratory was not fully open, so TEM testing cannot be performed. According to the data available in this article and the published literature, it can still be proved that Ag2O, NiO and CuFe2O4 grow together to form a heterojunction.

1) XRD data shows that the material we prepared contains Ag2O, NiO and CuFe2O4 three components.

2) After five cycles of recycling, the degradation efficiency of Ag2O-NiO/CuFe2O4 to MB was still as high as 94.02%. This phenomenon indirectly prove that Ag2O-NiO/CuFe2O4 was very stable, that is, Ag2O, NiO and CuFe2O4 is compounded together.

3)In this study, the method used to prepare heterogeneous catalyst is similar to many literatures, such as i)Samireh Mohammadi Aydoghmish, S.A. Hassanzadeh-Tabrizi, A. Saffar-Teluri. Facile synthesis and investigation of NiO–ZnO–Ag nanocomposites as efficient photocatalysts for degradation of methylene blue dye. Ceramics International, 2019,45:14934-14942. ii)Mokhtar Mohamed Mohamed, M. Khairy, Ahmed Ibrahem. P-n junction based Ag2O@Ag@Coated functionalized carbon nanotubes and their efficient visible-light photocatalytic reduction performances. Microporous and Mesoporous Materials,2020,292:109734.

Therefore, we infer that Ag2O-NiO/CuFe2O4 was successfully prepared.

Comment 8:

The material is coded as Ag2O@CuFe2O4@NiO, but “@” generally describes encapsulation of one material into another one. If Ag2O and NiO are on CuFe2O4, then another code such as Ag2O-NiO/CuFe2O4 is more suitable.

Response to reviewer 2:

Thank you very much for your advice. Ag2O@CuFe2O4@NiO has been instead of Ag2O-NiO/CuFe2O4 in the text.

Comment 9:

Figure captions must specify all experimental conditions. For example, is MB concentration 10 mg/L in Fig. 5 and 100 mg/L in Figure 6, as mentioned in the text? Is Fig 4 data for sunlight irradiations with H2O2 or without?

Response to reviewer 2:

I am sorry for my carelessness. In section 3.2, 100mg/l has been replaced by 10 mg/L. Fig 4 describes the catalytic performance of catalyst with and without H2O2 under sunlight irradiation.

Comment 10:

Lines 178-9 describes operation with H2O2, please specify.

Response to reviewer 2:

Thank you very much for your advice. In degradation experiment, 100 µL30% H2O2 was put into MB solution together with photocatalytic material. This detail was mentioned in section 2.1 and section 3.2.

Comment 11:

TOF must be defined and calculated.

Response to reviewer 2:

Thank you very much for this comment. After careful research, I think the TOF interpretation of this part of experimental data is improper, so it should be rewritten. The following is the rewritten part.

When the catalyst dosage is less than 10 mg, the catalytic efficiency increases with the catalyst dosage. Then, with the further increase of the catalyst dosage, the efficiency decreases significantly.

Comment 12:

The text contains erroneous terms and typos and English is poor; major editing is needed. Examples:

Abstract: the terms photofentun and ternyl do not exist.

Experimental: what is AR? Silver nitrate or silver sulfate? Sodium hydroxide, NOT sodium oxide. Methylene blue, NOT methyl blue.

Line 89: powder, NOT power.

Line 148, 100 µL, NOT 100 mL

Lines 186, 198, 199, µL, NOT uL.

Response to reviewer 2:

I am sorry for my carelessness. Now, I have taken a in-depth revision of the language for our manuscript.

“photofentun” has been replaced by “photo-Fenton”. “ternyl” has been replaced by “ternary”.

AR is the abbreviation of analytical reagent. The complete form of the analytical pure reagent has been supplemented in the text. In this experiment silver sulfate has been used , and the corresponding content has been modified. “Sodium oxide” has been replaced by “Sodium hydroxide”. “Methyl blue” has been replaced by “Methylene blue”.

“Power” has been replaced by “powder”.

“100 mL” has been replaced by “100 µL”

“uL” has been replaced by “µL”.

Thank you very much again for your comments. I hope this revision is satisfied to you and our paper could be accepted soon.

Reviewer 3 Report

The authors have used the hydrothermal method to prepare the
Ag2O@CuFe2O4@NiO ternary catalyst. It applied to the photo-Fenton reaction to degrade methylene blue. Methylene blue is a dye causing the problem of water pollution. 

There are inaccuracies in the text such as

  1. in line 83 "Appropriate amount of medicine NaON"... probably, the authors have meant NaOH. In line 88 " which is power C." it should be corrected to powder, and further in the text, such minor inaccuracies should be corrected.
  2. what are the degradation products of methylene blue?
    Has the degradation of methylene blue proceeded to ending products?
    In Figure 8, Degradation products are H2O and CO2 and .... 
    How was the composition of products in the gas and liquid phases evaluated?

Author Response

Dear reviewer 3,

Thank you very much for reviewing our manuscript, and finding out so detailed  mistakes which I think actually can improve the quality of our paper. I am also particularly grateful to you. Now, I have revised our manuscript according to your comments, all the revisions are highlighted in red, and the detailed revision processes are described as follows:

Comment 1:

in line 83 "Appropriate amount of medicine NaON"... probably, the authors have meant NaOH. In line 88 " which is power C." it should be corrected to powder, and further in the text, such minor inaccuracies should be corrected.

Response to reviewer 3:

I am sorry for my carelessness. “NaON” has been replaced by “NaOH”. “Power” has been replaced by “powder”. In the text minor inaccuracies have been modified.

Comment 2:

what are the degradation products of methylene blue?

Has the degradation of methylene blue proceeded to ending products?

In Figure 8, Degradation products are H2O and CO2 and .... 

How was the composition of products in the gas and liquid phases evaluated?

Response to reviewer 3:

The TOC test result shows that the MB mineralization rate was 78.64%, that is, 78.64% MB was decomposed into CO2. This study focuses on the mechanism of Ag2O-NiO/CuFe2O4 photo-Fenton degrade MB.  This study focuses on the mechanism of Ag2O-NiO/CuFe2O4 photo-Fenton degrade MB, so it is not evalutated gas and liquid phases. In future studies, the composition of products in the gas and liquid phases will be evaluated.

Thank you very much again for your comments. I hope this revision is satisfied to you and our paper could be accepted soon.

Round 2

Reviewer 2 Report

Despite noticeable improvement, serious concerns remain regarding synthetic protocols and characterisation of the materials. The authors should confirm more convincingly that they have obtained the desired title materials by identifying their key components, i.e. CuFe2O4, Ag2O and NiO. Otherwise, the manuscript cannot be accepted for publication. Comments:

  1. Synthetic protocols.
    • Lines 86-87: The authors have not clarified how the solvent (water) was removed at room temperature. Was this done at reduced pressure? Washing with water, as suggested, is not a method to remove water; this makes no sense at all.
    • Lines 91-94: Is the solids were separated by centrifugation-washing, how much Ag and Ni was expected to remain in the materials (and why) and how much was washed away? Nominal % of Ag and Ni should be compared to the actual content.
    • Line 94: The sentence “Put C into a muffle…” is syntactically incorrect. English should be thoroughly revised there and throughout the entire text.
    • Line 84: What is “medicine NaOH”?
    • pH, NOT PH
  2. Reference 33 is wrong. It should be replaced with: A.V. Puga, Photocatalytic production Photocatalytic production of hydrogen from biomass-derived feedstocks, Coord. Chem. Rev. 2016, 315, 1-66.
  3. The formation of CuFe2O4 is dubious. The XRD diffraction peaks do not coincide with other literature reports, e.g. J. Magnetism Magnet. Mater, 2017, 434, 30. This aspect is critical and must be unambiguously clarified by comparing the experimental results with the expected diffraction pattern from crystallographic databases. Otherwise, the entire paper would be invalidated.
  4. In line with the previous comment, the formation of Ag2O and NiO phases should be more convincingly proven by comparing with crystallographic database and literature data.
  5. XRD patterns of CuFe2O4, Ag2O/CuFe2O4 and NiO/CuFe2O4 must be shown in comparison to the ternary material.
  6. Material coding is inconsistent. If the final proposed code is “Ag2O-NiO/CuFe2O4”, then correct the following: “Ag2O@CuFe2O4@NiO” in Section 2.2. title; “Ag2O/CuFe2O4/NiO” in Abstract and in the captions of Figures 1, 4 and 6.
  7. The title states facts which are not proven: The presence and location of Ag2O and NiO are unclear.
  8. Figure captions must specify all experimental conditions. For example, MB and H2O2 concentrations in Figs. 4-6 must be specified.
  9. Equation labels (3)-(5) are now repeated (page 8). Please correct.
  10. The comparative performance of Ag2O/CuFe2O4 and NiO/CuFe2O4 must be commented in more detail (lines 121-2), and the corresponding plots included in the main text.

Author Response

Dear reviewer,

Thank you very very much for giving us so detailed and wonderful comments which actually can improve the quality of our manuscript. Now, we had carefully revised our manuscript point-by-point according to your comments. The detialed revisions are descibed as follow:

Comment 1:

Synthetic protocols.

Comment 1-1: Lines 86-87: The authors have not clarified how the solvent (water) was removed at room temperature. Was this done at reduced pressure? Washing with water, as suggested, is not a method to remove water; this makes no sense at all.

Response to reviewer:

I am sorry for our careless. The solvent (water) was not removed at room temperature, but at 60℃ in a common drying oven. I had revised this in the revised manuscript.

Comment 1-2: Lines 91-94: Is the solids were separated by centrifugation-washing, how much Ag and Ni was expected to remain in the materials (and why) and how much was washed away? Nominal % of Ag and Ni should be compared to the actual content.

Response to reviewer:

I am sorry that the statement for Ag2O-NiO/CuFe2O4 preparation was not clear in our previous manuscript. In fact, Ag2SO4, nickel acetate and powder C (CuFe-LDH) were mixed uniformly in deionzed water, then the mixture was dried at 60℃ in a common drying oven. Next, the solid mixture was calcined at 650℃ in a muffle furnace, harvesting the final photo-Fenton catalyst Ag2O-NiO/CuFe2O4. Therefore, the precursor and the product of Ag2O-NiO/CuFe2O4 were not washed with any solvent, implied that no Ag or Ni was lost. In addition, the ratio among Ag2O, NiO, CuFe2O4, had been optimized in our previous study. In this study, we just want to report a specific photo-Fenton catalyst posing excellent photocatalytic performance. So, the material ratio was not discussed in our manuscript, and we showed the specific dosage of chemicals in the chapter of “materials and methods”. In fact, the amount of both Ag2O and NiO was 10% of CuFe2O4 (molar ratio), and the ratio between Ag2O and NiO was 1:3 (molar ratio).

Comment 1-3: Line 94: The sentence “Put C into a muffle…” is syntactically incorrect. English should be thoroughly revised there and throughout the entire text.

Response to reviewer:

Thank you very much this comment. The whole “2.2. Synthesis and characterization of Ag2O-CuFe2O4/NiO” is rewritten and checked carefully in this revision. Moreover, the English language of our whole manuscript is improved by the co-author Yonglei An.

Comment 1-4: Line 84: What is “medicine NaOH”?

Response to reviewer:

I am sorry for my poor English. “medicine” has been deleted in revised manuscript.

Comment 1-5: pH, NOT PH

Response to reviewer:

I am sorry for my careless. All “PH” in the text and figures of whole manuscript have been changed into “pH”.

Comment 2:

Reference 33 is wrong. It should be replaced with: A.V. Puga, Photocatalytic production Photocatalytic production of hydrogen from biomass-derived feedstocks, Coord. Chem. Rev. 2016, 315, 1-66.

Response to reviewer:

I am sorry for my careless. I have replaced the literature in the revised manuscript.

Comment 3:

The formation of CuFe2O4 is dubious. The XRD diffraction peaks do not coincide with other literature reports, e.g. J. Magnetism Magnet. Mater, 2017, 434, 30. This aspect is critical and must be unambiguously clarified by comparing the experimental results with the expected diffraction pattern from crystallographic databases. Otherwise, the entire paper would be invalidated.

Response to reviewer:

Thank you very much for this comment. In fact, in our study, the diffraction peaks of as-prepared CuFe2O4 in XRD pattern can commendably match the standard pattern of CuFe2O4 (PDF card No. 72-1174), indicative of successful preparation on CuFe2O4. Of course, there may be some difference between our CuFe2O4 and other CuFe2O4 reported in literature, due to different experimental conditions. Moreover, the HRTEM results in our manuscript also found the 0.48 nm, 0.249 nm and 0.205 nm lattice plane spacings, corresponding to (101), (211) and (220) crystal planes of CuFe2O4 (PDF card No.72-1174), furtherly proving that it is CuFe2O4. In fact, CuFe2O4 is very simple material and easy to synthesized. Therefore, based on the XRD and HRTEM results, we are very confident that it undoubtedly is CuFe2O4.

Comment 4:

In line with the previous comment, the formation of Ag2O and NiO phases should be more convincingly proven by comparing with crystallographic database and literature data.

Response to reviewer:

Thank you very much for this comment. Actually, it is important to verify the existence of Ag2O and NiO phases in our as-prepared Ag2O-CuFe2O4/NiO. In this revision, the XRD pattern of CuFe2O4 is added in the manuscript. After comparing the XRD patterns of CuFe2O4 and Ag2O-CuFe2O4/NiO, the 32.74° and 53.75° diffraction peaks corresponding to the (111) and (220) crystal planes of Ag2O (PDF card No.12-793) were found in Ag2O-NiO/CuFe2O4. Also, the 37.29° and 62.67° diffraction peaks attributed to the (101) and (110) crystal planes of NiO phase (PDF card No.44-1159) were found in Ag2O-NiO/CuFe2O4. These results can convincingly verify the existence of Ag2O and NiO in Ag2O-NiO/CuFe2O4.

Comment 5:

XRD patterns of CuFe2O4, Ag2O/CuFe2O4 and NiO/CuFe2O4 must be shown in comparison to the ternary material.

Response to reviewer:

Thank you very much for this comment. In this study, we just want to report a specific photo-Fenton catalyst posing excellent photocatalytic performance. Therefore, the material ratio was not discussed in our manuscript, and we showed the specific dosage of chemicals in the chapter of “materials and methods”. In fact, in our previous study, the total amount of Ag2O and NiO was invariably 10% of CuFe2O4 (molar ratio), then the ratio (molar ratio) between Ag2O and NiO was set as 1:0, 3:1, 1:1, 1:3, 0:1. Furtherly, we had synthesized these kinds of photo-Fenton catalyst, then photo-Fenton experiments were carried out firstly. We found that when the molar ratio between Ag2O and NiO was 3:1, Ag2O-NiO/CuFe2O4 showed the best performance on MB photo-Fenton degradation. Therefore, we only characterized the Ag2O-NiO/CuFe2O4 with 3:1 molar ratio of Ag2O and NiO in our manuscript. Of course, the CuFe2O4 was also characterized as the control. Therefore, the XRD pattern of CuFe2O4 was added in the revised manuscript.

Comment 6:

Material coding is inconsistent. If the final proposed code is “Ag2O-NiO/CuFe2O4”, then correct the following: “Ag2O@CuFe2O4@NiO” in Section 2.2. title; “Ag2O/CuFe2O4/NiO” in Abstract and in the captions of Figures 1, 4 and 6.

Response to reviewer:

I am sorry for my careless. The code of our catalyst has been uniformly revised as “Ag2O-NiO/CuFe2O4” in the whole manuscript.

Comment 7:

The title states facts which are not proven: The presence and location of Ag2O and NiO are unclear.

Response to reviewer:

Thank you very much for this comment. This comment is similar to comment 4. Just like the response to comment 4, the XRD patterns of CuFe2O4 and Ag2O-NiO/CuFe2O4 can adequately prove the presence of Ag2O and NiO. However, it was a pity that we didn’t find the diffraction lattice fringe of Ag2O and NiO in the HRTEM, due to their low content and low crystallinity. Therefore, we cannot directly prove the location of Ag2O and NiO. Fortunately, the photocatalytic performance of Ag2O-NiO/CuFe2O4 on MB degradation was obviously higher than CuFe2O4, indicative of effective hybridization among Ag2O, NiO and CuFe2O4.

Comment 8:

Figure captions must specify all experimental conditions. For example, MB and H2O2 concentrations in Figs. 4-6 must be specified.

Response to reviewer:

Thank you very much for this good suggestion. I have supplemented the MB and H2O2 concentrations in the Figure captions in Figure 4, 5, 6 and 7 as follow:

Initial MB concentration: 10 mg/L. Initial H2O2 concentration: 416 mg/L.

Comment 9:

Equation labels (3)-(5) are now repeated (page 8). Please correct.

Response to reviewer:

I am sorry for my careless. Now, I have corrected all the photo-Fenton mechanism formulas in the revised manuscript.

Comment 10:

The comparative performance of Ag2O/CuFe2O4 and NiO/CuFe2O4 must be commented in more detail (lines 121-2), and the corresponding plots included in the main text.

Response to reviewer:

Thank you very much for this comment. As previously mentioned in the response to comment 5, we just wanted to report a specific kind of photo-Fenton catalyst Ag2O-NiO/CuFe2O4. For Ag2O/CuFe2O4 and NiO/CuFe2O4, we had verified their weaker photocatalytic activties than Ag2O-NiO/CuFe2O4 in our previous study. Therefore, Ag2O/CuFe2O4 and NiO/CuFe2O4 were not important in our study. The aim of this study was to investigate the photo-Fenton catalytic performance and mechanism of Ag2O-NiO/CuFe2O4. Of course, to investigate Ag2O/CuFe2O4 and NiO/CuFe2O4 would contribute to the understanding of ternary catalyst Ag2O-NiO/CuFe2O4, which will be carried out in our future study.

These are all our responses to your comments. The corresponding revison has been done carefully in our manuscript.

Thank you very much again for giving us so wonderful and valuable comments which actually improve the quality of our manuscript effectively.

I sincerely hope that our responses and revisions are satisfied to you and our manuscript could be accepted soon.

Kind regards,

Lu Liu
